# ADAPTIVE SELF-TRAINING FOR NEURAL SEQUENCE LABELING WITH FEW LABELS

## ABSTRACT

Neural sequence labeling is an important technique employed for many Natural Language Processing (NLP) tasks, such as Named Entity Recognition (NER), slot tagging for dialog systems and semantic parsing. Large-scale pre-trained language models obtain very good performance on these tasks when fine-tuned on large amounts of task-specific labeled data. However, such large-scale labeled datasets are difficult to obtain for several tasks and domains due to the high cost of human annotation as well as privacy and data access constraints for sensitive user applications. This is exacerbated for sequence labeling tasks requiring such annotations at token-level. In this work, we develop techniques to address the label scarcity challenge for neural sequence labeling models. Specifically, we develop self-training and meta-learning techniques for training neural sequence taggers with few labels. While self-training serves as an effective mechanism to learn from large amounts of unlabeled data – meta-learning helps in adaptive sample re-weighting to mitigate error propagation from noisy pseudo-labels. Extensive experiments on six benchmark datasets including two for massive multilingual NER and four slot tagging datasets for task-oriented dialog systems demonstrate the effectiveness of our method. With only 10 labeled examples for each class for each task, our method obtains $10\%$ improvement over state-of-the-art systems demonstrating its effectiveness for the low-resource setting.

## 1 INTRODUCTION

**Motivation.** Deep neural networks typically require large amounts of training data to achieve state-of-the-art performance. Recent advances with pre-trained language models like BERT (Devlin et al., 2019), GPT-2 (Radford et al., 2019) and RoBERTa (Liu et al., 2019) have reduced this annotation bottleneck. In this paradigm, large neural network models are trained on massive amounts of unlabeled data in a self-supervised manner. However, the success of these large-scale models still relies on fine-tuning them on large amounts of labeled data for downstream tasks. For instance, our experiments show 27% relative improvement on an average when fine-tuning BERT with the full training set (2.5K-705K labels) vs. fine-tuning with only 10 labels per class. This poses several challenges for many real-world tasks. Not only is acquiring large amounts of labeled data for every task expensive and time consuming, but also not feasible in many cases due to data access and privacy constraints. This issue is exacerbated for sequence labeling tasks that require annotations at *token-* and *slot-level* as opposed to instance-level classification tasks. For example, an NER task can have slots like *B-PER, I-PER, O-PER* marking the beginning, intermediate and out-of-span markers for person names, and similar slots for the names of location and organization. Similarly, language understanding models for dialog systems rely on effective identification of what the user intends to do (*intents*) and the corresponding values as arguments (*slots*) for use by downstream applications. Therefore, fully supervised neural sequence taggers are expensive to train for such tasks, given the requirement of thousands of annotations for hundreds of slots for the many different intents.

Semi-supervised learning (SSL) (Chapelle et al., 2010) is one of the promising paradigms to address labeled data scarcity by making effective use of large amounts of unlabeled data in addition to task-specific labeled data. Self-training (ST, (III, 1965)) as one of the earliest SSL approaches has recently shown state-of-the-art performance for tasks like image classification (Li et al., 2019; Xie et al., 2020) performing at par with supervised systems while using very few training labels. In contrast to such instance-level classification tasks, sequence labeling tasks have dependencies

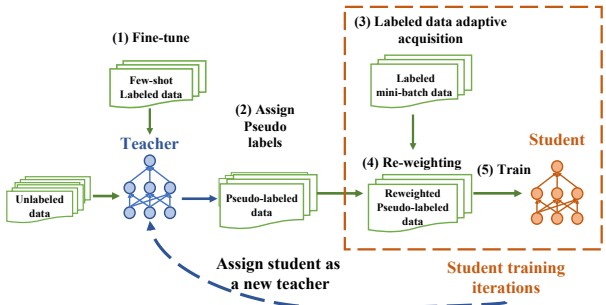

Figure 1: MetaST framework.

between the slots demanding different design choices for slot-level loss optimization for the limited labeled data setting. For instance, prior work (Ruder & Plank, 2018) using classic self-training techniques for sequence labeling did not find much success in the low-data regime with $10\%$ labeled data for the target domain. Although there has been some success with careful task-specific data selection (Petrov & McDonald, 2012) and more recently for distant supervision (Liang et al., 2020) using external resources like knowledge bases (e.g., Wikipedia). In contrast to these prior work, we develop techniques for self-training with limited labels and without any task-specific assumption or external knowledge.

For self-training, a base model (*teacher*) is trained on some amount of labeled data and used to pseudo-annotate (task-specific) unlabeled data. The original labeled data is augmented with the pseudo-labeled data and used to train a *student* model. The student-teacher training is repeated until convergence. Traditionally in self-training frameworks, the teacher model pseudo-annotates unlabeled data without any sample selection. This may result in gradual drifts from self-training on noisy pseudo-labeled instances (Zhang et al., 2017). In order to deal with noisy labels and training set biases, Ren et al. (2018) propose a meta-learning technique to automatically re-weight noisy samples by their loss changes on a held-out clean labeled validation set. We adopt a similar principle in our work and leverage *meta-learning* to re-weight noisy pseudo-labeled examples from the teacher. While prior techniques for learning to re-weight examples have been developed for instance-level classification tasks, we extend them to operate at token-level for discrete sequence labeling tasks. To this end, we address some key challenges on how to construct an informative held-out validation set for token-level re-weighting. Prior works (Ren et al., 2018; Shu et al., 2019) for instance classification construct this validation set by random sampling. However, sequence labeling tasks involve many slots (e.g. WikiAnn has 123 slots over 41 languages) with variable difficulty and distribution in the data. In case of random sampling, the model oversamples from the most populous category and slots. This is particularly detrimental for low-resource languages in the multilingual setting. To this end, we develop an *adaptive* mechanism to create the validation set on the fly considering the diversity and uncertainty of the model for different slot types. Furthermore, we leverage this validation set for token-level loss estimation and re-weighting pseudo-labeled sequences from the teacher in the meta-learning setup. While prior works (Li et al., 2019; Sun et al., 2019; Bansal et al., 2020) on meta-learning for image and text classification leverage *multi-task* learning to improve a target classification task based on several similar tasks, in this work we focus on a single sequence labeling task – making our setup more challenging altogether.

**Our task and framework overview.** We focus on sequence labeling tasks with only a few annotated samples (e.g., $K = \{5, 10, 20, 100\}$) per slot type for training and large amounts of task-specific unlabeled data. Figure 1 shows an overview of our framework with the following components: (i) *Self-training:* Our self-training framework leverages a pre-trained language model as a teacher and co-trains a student model with iterative knowledge exchange (ii) *Adaptive labeled data acquisition for validation:* Our few-shot learning setup assumes a small number of labeled training samples per slot type. The labeled data from multiple slot types are not equally informative for the student model to learn from. While prior works in meta-learning randomly sample some labeled examples for held-out validation set, we develop an adaptive mechanism to create this set on the fly. To this end, we leverage loss decay as a proxy for model uncertainty to select informative labeled samples for the student model to learn from in conjunction with the re-weighting mechanism in the next step. (iii) *Meta-learning for sample re-weighting:* Since pseudo-labeled samples from the teacher can be noisy, we employ meta-learning to re-weight them to improve the student model performance on the held-out validation set obtained from the previous step. In contrast to prior work (Ren et al., 2018)

on sample re-weighting operating at instance-level, we incorporate the re-weighting mechanism at *token-level* for sequence labeling tasks. Here the token-level weights are determined by the student model loss on the above validation set. Finally, we learn all of the above steps jointly with end-to-end learning in the self-training framework. We refer to our adaptive self-training framework with meta-learning based sample re-weighting mechanism as `MetaST`.

We perform extensive experiments on six benchmark datasets for several tasks including multi-lingual Named Entity Recognition and slot tagging for user utterances from task-oriented dialog systems to demonstrate the generalizability of our approach across diverse tasks and languages. We adopt BERT and multilingual BERT as encoder and show that its performance can be significantly improved by nearly 10% for low-resource settings with few training labels (e.g., 10 labeled examples per slot type) and large amounts of unlabeled data. In summary, our work makes the following contributions. (i) Develops a self-training framework for neural sequence tagging with few labeled training examples. (ii) Leverages an acquisition strategy to adaptively select a validation set from the labeled set for meta-learning of the student model. (iii) Develops a meta-learning framework for re-weighting pseudo-labeled samples at token-level to reduce drifts from noisy teacher predictions. (iv) Integrates the aforementioned components into an end-to-end learning framework and demonstrates its effectiveness for neural sequence labeling across six benchmark datasets with multiple slots, shots, domains and languages.

## 2 BACKGROUND

**Sequence labeling and slot tagging.** This is the task identifying the entity *span* of several slot types (e.g., names of person, organization, location, date, etc.) in a text sequence. Formally, given a sentence with $N$ tokens $X = \{x_1, ..., x_N\}$, an entity or slot value is a span of tokens $s = [x_i, ..., x_j](0 \leq i \leq j \leq N)$ associated with a type. This task assumes a pre-defined tagging policy like `BIO` (Tjong et al., 1999), where `B` marks the beginning of the slot, `I` marks an intermediate token in the span, and `O` marks out-of-span tokens. These span markers are used to extract multi-token values for each of the slot types with phrase-level evaluation for the performance.

**Self-training.** Consider $f(\cdot; \theta_{tea})$ and $f(\cdot; \theta_{stu})$ to denote the teacher and student models respectively in the self-training framework. The role of the teacher model (e.g., a pre-trained language model) is to assign pseudo-labels to unlabeled data that is used to train a student model. The teacher and student model can exchange knowledge and the training schedules are repeated till convergence. The success of self-training with deep neural networks in recent works (He et al., 2019; Xie et al., 2020) has been attributed to a number of factors including stochastic regularization with dropouts and data regularization with unlabeled data. Formally, given $m$-th unlabeled sentence with $N$ tokens $X_m^u = \{x_{1,m}^u, ..., x_{N,m}^u\}$ and $C$ pre-defined labels, consider the pseudo-labels $\hat{Y}_m^{(t)} = [\hat{y}_{m,1}^{(t)}, ..., \hat{y}_{m,N}^{(t)}]$ generated by the teacher model at the $t$-th iteration where,

$$\hat{y}_{m,n}^{(t)} = \arg\max_{c \in C} f_{n,c}(x_{m,n}^u; \theta_{tea}^{(t)}). \tag{1}$$

The pseudo-labeled data set, denoted as $(X^u, \hat{Y}^{(t)}) = \{(X_m^u, \hat{Y}_m^{(t)})\}_m^M$, is used to train the student model and learn its parameters as:

$$\hat{\theta}_{stu}^{(t)} = \arg\min_{\theta} \frac{1}{M} \sum_{m=1}^{M} l(\hat{Y}_m^{(t)}, f(X_m^u; \theta_{stu}^{(t-1)})), \tag{2}$$

where $l(\cdot, \cdot)$ can be modeled as the cross-entropy loss.

## 3 ADAPTIVE SELF TRAINING

Given a pre-trained language model (e.g., BERT (Devlin et al., 2019)) as the teacher, we first fine-tune it on the small labeled data to make it aware of the underlying task. The fine-tuned teacher model is now used to pseudo-label the large unlabeled data. We consider the student model as another instantiation of the pre-trained language model that is trained over the pseudo-labeled data. However, our few-shot setting with limited labeled data results in a noisy teacher. A naive transfer of teacher knowledge to the student results in the propagation of noisy labels limiting the performance of the student model. To address this challenge, we develop an *adaptive* self-training framework to

re-weight pseudo-labeled predictions from the teacher with a meta-learning objective that optimizes the token-level loss from the student model on a held-out labeled validation set. This held-out set is adaptively constructed via labeled data acquisition which selects labeled samples with high uncertainty for efficient data exploration.

### 3.1 ADAPTIVE LABELED DATA ACQUISITION

In standard meta-learning setup for instance-level classification tasks, the held-out validation set is usually constructed via random sampling (Ren et al., 2018; Shu et al., 2019). Sequence labeling tasks involve many slot types with variable difficulty and distribution in the data. For instance, NER tasks over WikiAnn operate over 123 slot types from 41 languages with additional complexity from variable model performance across different languages. A random sampling leads to oversampling instances with the most populous categories and slot types in the data. Therefore, we propose a novel labeled data acquisition strategy to construct the validation set for effective data exploration. We demonstrate its benefit over classic meta-learning approaches from prior works in experiments.

In general, data acquisition strategies for prior works in meta-learning and active learning broadly leverage random sampling (Ren et al., 2018; Shu et al., 2019), easy (Kumar et al., 2010) and hard example mining (Shrivastava et al., 2016) or uncertainty-based methods (Chang et al., 2017a). These strategies have been compared in prior works (Chang et al., 2017a; Gal et al., 2017) that show uncertainty-based methods to have better generalizability across diverse settings. There are several approaches to uncertainty estimation including error decay (Konyushkova et al., 2017; Chang et al., 2020), Monte Carlo dropouts (Gal et al., 2017) and predictive variance (Chang et al., 2017a). We follow a similar principle of error decay to find samples that the model is uncertain about and can correspondingly benefit from knowing their labels (similar to active learning settings). To this end, we leverage stochastic loss decay from the model as a proxy for the model uncertainty to generate validation set on the fly. This is used for estimating token-level weights and re-weighting pseudo labeled data in Section 3.2.

Consider the loss of the student model with parameters $\theta_{stu}^{(t)}$ on the labeled data $(X_m^l, Y_m)$ in the $t$-th iteration as $l(Y_m, f(X_m^l; \theta_{stu}^{(t)}))$. To measure the loss decay value at any iteration, we use the difference between the current and previous loss values. Considering these values may fluctuate across iterations, we adopt the moving average of the loss values for $(X_m^l, Y_m)$ in the latest $R$ iterations as a baseline $l_b^m$ for loss decay estimation. Baseline measure $l_b^m$ is calculated as follows:

$$l_b^m = \frac{1}{R} \sum_{r=1}^{R} l(Y_m, f(X_m^l; \theta_{stu}^{(t-r)})).$$ (3)

Since the loss decay values are estimated on the fly, we want to balance exploration and exploitation. To this end, we add a smoothness factor $\delta$ to prevent the low loss decay samples (i.e. samples with low uncertainty) from never being selected again. Considering all of the above factors, we obtain the sampling weight of labeled data $(X_m^l, Y_m^l)$ as follows:

$$W_m \propto \max(l_b^m - l(Y_m, f(X_m^l; \theta_{stu}^{(t)})), 0) + \delta.$$ (4)

The smoothness factor $\delta$ needs to be adaptive since the training loss is dynamic. Therefore, We adopt the maximum of the loss decay value as the smoothness factor $\delta$ to encourage exploration.

The aforementioned acquisition function is re-estimated after a fixed number of steps to adapt to model changes. With labeled data acquisition, we rely on informative uncertain samples to improve learning efficiency. The sampled mini-batches of labeled data $\{\mathcal{B}_s^l\}$ are used as a validation set for the student model in the next step for re-weighting pseudo-labeled data from the teacher model. We demonstrate its impact via ablation study in experiments. Note that the labeled data is only used to compute the acquisition function and not used for explicit training of the student model in this step.

### 3.2 RE-WEIGHTING PSEUDO-LABELED DATA

To mitigate error propagation from noisy pseudo-labeled sequences from the teacher, we leverage meta-learning to adaptively re-weight them based on the student model loss on the held-out validation set obtained via labeled data acquisition from the previous section. In contrast to prior work focusing on instance-level tasks like image classification – sequence labeling operates on discrete text sequences as input and assigns labels to each token in the sequence. Since teacher predictions vary for different slot labels and types, we adapt the meta-learning framework to re-weight samples at a token-level resolution.

**Token Re-weighting.** Consider the pseudo-labels $\{\hat{Y}_m^{(t)} = [\hat{y}_{m,1}^{(t)}, ..., \hat{y}_{m,N}^{(t)}]\}_{m=1}^M$ from the teacher in the $t$-th iteration with $m$ and $n$ indexing the instance and a token in the instance, respectively. In classic self-training, we update the student parameters leveraging pseudo-labels as follows:

$$\hat{\theta}_{stu}^{(t)} = \hat{\theta}_{stu}^{(t-1)} - \alpha\nabla\big(\frac{1}{M}\sum_{m=1}^M l(\hat{Y}_m^{(t)}, f(X_m^u; \theta_{stu}^{(t-1)}))\big). \tag{5}$$

Now, to downplay noisy token-level labels, we leverage meta-learning to re-weight the pseudo-labeled data. To this end, we follow a similar analysis from (Koh & Liang, 2017) and (Ren et al., 2018) to perturb the weight for each token in the mini-batch by $\epsilon$. Weight perturbation is used to discover data points that are most important to improve the model performance on a held-out validation set (Koh & Liang, 2017) where the sample importance is given by the magnitude of the the negative gradients. We extend prior techniques to obtain token-level perturbations as:

$$\hat{\theta}_{stu}^{(t)}(\epsilon) = \hat{\theta}_{stu}^{(t-1)} - \alpha\nabla\big(\frac{1}{M}\frac{1}{N}\sum_{m=1}^M\sum_{n=1}^N[\epsilon_{m,n} \cdot l(\hat{y}_{m,n}^{(t)}, f(x_{m,n}^u; \hat{\theta}_{stu}^{(t-1)}))]\big). \tag{6}$$

The token weights are obtained by minimizing the student model loss on the held-out validation set. Here, we employ the labeled data acquisition strategy from Eq. 4 to sample informative mini-batches of labeled data $\mathcal{B}_s^l$ locally at step $t$. To obtain a cheap estimate of the meta-weight at step $t$, we take a single gradient descent step for the sampled labeled mini-batch $\mathcal{B}_s^l$ :

$$u_{m,n,s} = -\frac{\partial}{\partial\epsilon_{m,n,s}}\big(\frac{1}{|\mathcal{B}_s^l|}\frac{1}{N}\sum_{m=1}^{|\mathcal{B}_s^l|}\sum_{n=1}^N[l(y_{m,n}, f(x_{m,n}^l; \hat{\theta}_{stu}^{(t)}(\epsilon)))]\big)|_{\epsilon_{m,n,s}=0} \tag{7}$$

We set the token weights to be proportional to the negative gradients to reflect the importance of pseudo-labeled tokens in the sequence. Since sequence labeling tasks have dependencies between the slot types and tokens, it is difficult to obtain a good estimation of the weights based on a single mini-batch of examples. Therefore, we sample $S$ mini-batches of labeled data $\{\mathcal{B}_1^l, ..., \mathcal{B}_S^l\}$ with the adaptive acquisition strategy and calculate the mean of the gradients to obtain a robust gradient estimate. Note that $S$ is a constant number that is the same for each token and the proportional sign in Eq. 8. Since a negative weight indicates a pseudo-label of poor quality that would potentially degrade the model performance, we set such weights to 0 to filter them out. The impact of $S$ is investigated in the experiments (refer to Appendix A.1). The overall meta-weight of pseudo-labeled token $(x_{m,n}^u, \hat{y}_{m,n})$ is obtained as:

$$w_{m,n} \propto \max(\sum_{s=1}^S u_{m,n,s}, 0) \tag{8}$$

To further ensure the stability of the loss function in each mini-batch, we normalise the weight $w_{m,n}$. Finally, we update the student model parameters while accounting for token-level re-weighting as:

$$\hat{\theta}_{stu}^{(t)} = \hat{\theta}_{stu}^{(t-1)} - \alpha\nabla\big(\frac{1}{M}\frac{1}{N}\sum_{m=1}^M\sum_{n=1}^N[w_{m,n} \cdot l(\hat{y}_{m,n}^{(t)}, f(x_{m,n}^u; \hat{\theta}_{stu}^{(t-1)}))]\big). \tag{9}$$

We demonstrate the impact of our re-weighting mechanism with an ablation study in experiments.

### 3.3 TEACHER MODEL ITERATIVE UPDATES

At the end of every self-training iteration, we assign the student model as a new teacher model (i.e., $\theta_{tea} = \theta_{stu}^{(T)}$) . Since the student model uses the labeled data only as a held-out validation set for meta-learning, we further utilize the labeled data $(X^l, Y)$ to fine-tune the new teacher model

$f(\cdot, \theta_{tea}^{(t)})$ with standard supervised loss minimization. We explore the effectiveness of this step with an ablation study in experiments. The overall training procedure is summarized in Algorithm 1.

---

**Algorithm 1:** MetaST Algorithm.

---

**Input:** Labeled sequences $(X^l, Y)$; Unlabeled sequences $(X^u)$; Pre-trained BERT model with randomly initialized token classification layer $f(\cdot; \theta^{(0)})$; Batches $S$; Number of self-training iterations $T$.

Initialize teacher model $\theta_{tea} = \theta^{(0)}$

**while** *not converged* **do**

    Fine-tune teacher model on small labeled data $(X^l, Y)$;

    Initialize the student model $\theta_{stu}^{(0)} = \theta^{(0)}$;

    Generate hard pseudo-labels $\hat{Y}^{(t)}$ for unlabeled samples $X^u$ with model $f(\cdot, \theta_{tea})$;

    **for** $t \leftarrow 1$ **to** $T$ **do**

        Compute labeled data acquisition function according to Eq. 4;

        Sample $S$ mini-batches of labeled examples $\{\mathcal{B}_1^l, ..., \mathcal{B}_S^l\}$ from $(X^l, Y)$ based on labeled data acquisition function;

        Randomly sample a batch of pseudo-labeled examples $\mathcal{B}_u$ from $(X^u, \hat{Y}^{(t)})$ ;

        Compute token-level weights in $\mathcal{B}_u$ based on the loss on $\{\mathcal{B}_1^l, ..., \mathcal{B}_S^l\}$ according to Eq. 8;

        Train model $f(\cdot, \theta_{stu}^{(t)})$ on weighted pseudo-labeled sequences $\mathcal{B}_u$ and update parameters $\theta_{stu}^{(t)}$ ;

    **end**

    Update the teacher: $\theta_{tea} = \theta_{stu}^{(T)}$

**end**

---

## 4 EXPERIMENTS

**Encoder.** Pre-trained language models like BERT (Devlin et al., 2019), GPT-2 (Radford et al., 2019) and RoBERTa (Liu et al., 2019) have shown state-of-the-art performance for various natural language processing tasks. In this work we adopt one of them as a base encoder by initializing the teacher with pre-trained BERT-base model and a randomly initialized token classification layer.

**Datasets.** We perform large-scale experiments with six different datasets including user utterances for task-oriented dialog systems and multilingual Named Entity Recognition tasks as summarized in Table 1. *(a) Email*. This consists of natural language user utterances for email-oriented user actions like sending, receiving or searching emails with attributes like date, time, topics, people, etc. *(b) SNIPS* is a public benchmark

| Dataset | # Slots | # Train | # Test | # Lang |
|---|---|---|---|---|
| Email | 20 | 2.5K | 1k | EN |
| SNIPS | 39 | 13K | 0.7K | EN |
| MIT Movie | 12 | 8.8K | 2.4K | EN |
| MIT Restaurant | 8 | 6.9K | 1.5K | EN |
| Wikiann (EN) | 3 | 20K | 10K | EN |
| CoNLL03 (EN) | 4 | 15K | 3.6K | EN |
| CoNLL03 | 16 | 38K | 15K | 4 |
| Wikiann | 123 | 705K | 329K | 41 |

Table 1: Dataset summary.

dataset (Coucke et al., 2018) of user queries from multiple domains including music, media, and weather. *(c) MIT Movie and Restaurant* corpus (Liu et al., 2013) consist of similar user utterances for movie and restaurant domains. (d) CoNLL03 (Sang & Meulder, 2003) and Wikiann (Pan et al., 2017) are public benchmark datasets for multilingual Named Entity Recognition. CoNLL03 is a collection of news wire articles from the Reuters Corpus from 4 languages with manual annotations, whereas Wikiann comprises of extractions from Wikipedia articles from 41 languages with automatic annotation leveraging meta-data for different entity types like ORG, PER, LOC etc. For every dataset, we sample $K \in \{5, 10, 20, 100\}$ labeled sequences for each slot type from the Train data, and add the remaining to the unlabeled set while ignoring their labels – following standard setups for semi-supervised learning. We repeatedly sample $K$ labeled instances three times for multiple runs to report average performance with standard deviation across the runs.

**Baselines.** The first baseline we consider is the fully supervised BERT model trained on all available training data which provides the ceiling performance for every task. Each of the other models are trained on $K$ training labels per slot type. We adopt several state-of-the-art semi-supervised methods as baselines: (1) CVT (Clark et al., 2018) is a semi-supervised sequence labeling method based on cross-view training; (2) SeqVAT (Chen et al., 2020) incorporates adversarial training with conditional random field layer for semi-supervised sequence labeling; (3) Mean Teacher (MT) (Tarvainen & Valpola, 2017) averages model weights to obtain an aggregated teacher; (4) VAT (Miyato et al., 2018) adopts virtual adversarial training to make the model robust to noise; (5) classic ST (III, 1965) is simple self-training method with hard pseudo-labels; (6) BOND (Liang et al., 2020) is the most recent work on self-training for sequence labeling with confidence-based sample selection and forms a strong baseline for our work. We implement our framework in Pytorch and use Tesla V100 gpus for experiments. Hyper-parameter configurations with model settings presented in Appendix.

**Neural sequence labeling performance with few training labels.** Table 2 shows the performance comparison among different models with K=10 labeled examples per slot type. The fully supervised BERT trained on thousands of labeled examples provides the ceiling performance for the few-shot setting. We observe our method MetaST to significantly outperform all methods across all datasets including the models that also use the same BERT encoder as ours like MT, VAT, Classic ST and BOND with corresponding average performance improvements as $14.22\%$, $14.90\%$, $8.46\%$ and $8.82\%$. Non BERT models like CVT and SeqVAT are consistently worse than other baselines.

| Method | SNIPS | Email | Movie | Restaurant | CoNLL03 (EN) | Wikiann (EN) |
|---|---|---|---|---|---|---|
| **# Slots** | 39 | 20 | 12 | 8 | 4 | 3 |
| **Full-supervision** | | | | | | |
| BERT | 95.80 | 94.44 | 87.87 | 78.95 | 92.40 | 84.04 |
| **Few-shot supervision (10 labels per slot)** | | | | | | |
| BERT | 79.01 | 87.85 | 69.50 | 54.06 | 71.15 | 45.61 |
| **Few-shot supervision (10 labels per slot) + unlabeled data** | | | | | | |
| CVT | 78.23 | 78.24 | 62.73 | 42.57 | 54.31 | 27.89 |
| SeqVAT | 78.67 | 72.65 | 67.10 | 51.55 | 67.21 | 35.16 |
| MT | 79.48 | 89.53 | 67.62 | 51.75 | 68.67 | 41.43 |
| VAT | 79.08 | 89.71 | 70.17 | 53.34 | 65.03 | 38.81 |
| Classic ST | 83.26 | 90.70 | 71.88 | 56.80 | 70.99 | 46.15 |
| BOND | 83.54 | 89.75 | 70.91 | 55.78 | 69.56 | 48.73 |
| MetaST | **88.23** (0.04;↑12%) | **92.18** (0.47;↑4.93%) | **77.67** (0.10;↑11.76%) | **63.83** (1.62;↑18.07%) | **76.65** (0.73;↑7.73%) | **56.61** (0.4;↑24.12%) |

Table 2: F1 score comparison of models for sequence labeling on different datasets. All models (except CVT and SeqVAT) use the same BERT encoder. F1 score of our model for each task is followed by standard deviation and percentage improvement (↑) over BERT with few-shot supervision.

We also observe variable performance of the models across different tasks. Specifically, the performance gap between the best few-shot model and the fully supervised model varies significantly. MetaST achieves close performance to the fully-supervised model in some datasets (e.g. SNIPS and Email) but has bigger room for improvement in others (e.g. CoNLL03 (EN) and Wikiann (EN)). This can be attributed to the following factors. (i) *Labeled training examples and slots*. The total number of labeled training instances for our K-shot setting is given by $K \times \#Slots$. Therefore, for tasks with higher number of slots and consequently more training labels, most of the models perform better including MetaST. Task-oriented dialog systems with more slots and inherent dependency between the slot types benefit more than NER tasks. (ii) *Task difficulty:* User utterances from task-oriented dialog systems for some of the domains like weather, music and emails contain predictive query patterns and limited diversity. In contrast, Named Entity Recognition datasets are comparatively diverse and require more training labels to generalize well. Similar observations are also depicted in Table 3 for multilingual NER tasks with more slots and consequently more training labels from multiple languages as well as richer interactions across the slots from different languages.

| Dataset | #Lang | #Slots | Full Sup. | Few-shot Sup. | \multicolumn{5}{c}{Few-shot supervision + unlabeled data} |
|---|---|---|---|---|---|---|---|---|---|
| | | | BERT | BERT | MT | VAT | Classic ST | BOND | MetaST |
| CoNLL03 | 4 | 16 | 87.67 | 70.77 | 68.34 | 67.63 | 72.69 | 72.79 | **76.41 (0.47) (↑ 7.97%)** |
| Wikiann | 41 | 123 | 87.17 | 79.67 | 80.23 | 78.82 | 80.24 | 79.57 | **81.61 (0.14) (↑ 2.42%)** |

Table 3: F1 score comparison of models for sequence labeling on multilingual datasets using the same BERT-Multilingual-Base encoder. F1 score of MetaST for each task is followed by standard deviation in parentheses and percentage improvement (↑) over BERT with few-shot supervision.

**Controlling for the total amount of labeled data.** In order to control for the variable amount of training labels across different datasets, we perform another experiment where we vary the number of labels for different slot types while keeping the total number of labeled instances for each dataset similar (ca. 200). Results are shown in Table 4. To better illustrate the effect of the number of training labels, we choose tasks with lower performance in Table 2 for this experiment. Comparing the results in Tables 2 and 4, we observe the performance of MetaST to improve with more training labels for all the tasks .

**Effect of varying the number of labels $K$ per slot.** Table 5 shows the improvement in the performance of MetaST when increasing the number of labels for each slot type in the SNIPS dataset.

| Dataset | BERT (Full Supervision) | BERT (Few-shot Supervision) | MetaST ( %Improvement ) |
|---|---|---|---|
| MIT Movie | 87.87 | 75.81 | **80.33** (↑ **5.96%**) |
| MIT Restaurant | 78.95 | 60.12 | **67.86** (↑ **12.87%**) |
| CoNLL03 (EN) | 92.40 | 77.48 | **81.61** (↑ **5.33%**) |
| Wikiann (EN) | 84.04 | 62.04 | **71.27** (↑ **14.88%**) |
| Average | 85.82 | 68.86 | **75.27** (↑ **9.31%**) |

Table 4: F1 scores of different models with 200 labeled samples for each task. The percentage improvement (↑) is over the BERT model with few-shot supervision.

Similar trends can be found on other datasets (results in Appendix). As we increase the amount of labeled training instances, the performance of BERT also improves, and correspondingly the margin between MetaST and these baselines decreases although MetaST still improves over all of them. In the self-training framework, given the ceiling performance for every task and the improved performance of the teacher with more training labels, there is less room for (relative) improvement of the student over the teacher model. Consider SNIPS for example. Our model obtains 12% and 2% improvement over the few-shot BERT model for the 10-shot and 100-shot setting with F1-scores as 88.22% and 95.39%, respectively. The ceiling performance for this task is 95.8% on training BERT on the entire dataset with 13K labeled examples. This demonstrates that MetaST is most impactful for low-resource settings with few training labels for a given task.

| #Slots | Few-shot Supervision | Few-shot supervision + unlabeled data | | | | | | |
|---|---|---|---|---|---|---|---|---|
| | BERT | CVT | SeqVAT | MT | VAT | Classic ST | BOND | MetaST (%Improvement) |
| 5 | 70.63 | 69.82 | 69.34 | 70.85 | 71.34 | 72.59 | 72.85 | **81.56** (↑15%) |
| 10 | 79.01 | 78.23 | 78.67 | 79.48 | 79.08 | 83.26 | 83.54 | **88.22** (↑12%) |
| 20 | 86.81 | 88.04 | 85.05 | 87.31 | 88.19 | 88.32 | 88.93 | **91.99** (↑6%) |
| 100 | 93.90 | 94.61 | 91.46 | 94.26 | 94.53 | 93.92 | 94.22 | **95.39** (↑2%) |

Table 5: Variation in model performance on varying $K$ labels / slot on SNIPS dataset with 39 slots. The percentage improvement (↑) is relative to the BERT model with few-shot supervision.

**Ablation analysis.** Table 6 demonstrates the impact of different MetaST components with ablation analysis. We observe that soft pseudo-labels hurt the model performance compared to hard pseudo-labels, as also shown in recent work (Kumar et al., 2020). Such a performance drop may be attributed to soft labels being less informative compared to sharpened ones. Removing the iterative teacher fine-tuning step (Section 3.1) also hurts the overall performance.

| Method | Datasets | |
|---|---|---|
| | SNIPS | CoNLL03 |
| BERT w/ Continued Pre-training + Few-shot Supervision | 83.96 | 69.84 |
| Classic ST | 83.26 | 70.99 |
| Classic ST w/ Soft Pseudo-Labels | 81.17 | 71.87 |
| MetaST (ours) w/ Hard Pseudo-Labels | **88.23** | **76.65** |
| MetaST w/ Soft Pseudo-Labels | 86.16 | 75.84 |
| MetaST w/o Iterative Teacher Fine-tune | 85.64 | 72.74 |
| MetaST w/o Labeled Data Acq. | 86.63 | 75.02 |
| **Pseudo-labeled Data Re-weighting** | | |
| MetaST w/o Re-weighting | 85.48 | 73.02 |
| MetaST (Easy) | 85.56 | 74.53 |
| MetaST (Difficult) | 86.34 | 68.06 |

Table 6: Ablation analysis of our framework MetaST with 10 labeled examples per slot on SNIPS and CoNLL03 (EN).

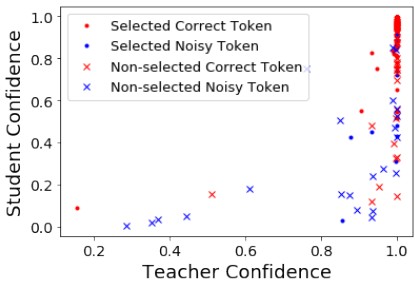

Figure 2: Visualization of MetaST re-weighting on CoNLL03 (EN).

**Continued pre-training v.s. self-training.** To contrast continued pre-training with self-training, we further pre-train BERT on in-domain unlabeled data and then fine-tune it with few labeled examples denoted as "BERT (Continued Pre-training + Few-shot Supervision)". The pre-training step improves the BERT performance over the baseline on SNIPS but degrades the performance on CoNLL03. This indicates that continued pre-training can improve the performance of few-shot supervised BERT on specialized tasks (e.g., SNIPS) with different data distribution than the original pre-training data (e.g., Wikipedia), but may not help for general domain ones like CoNLL03 with overlapping data from Wikipedia. In contrast to the above baseline, MetaST brings significant improvements on both datasets. This demonstrates the generality and flexibility of self-training over pre-training as also observed in contemporary work (Zoph et al., 2020) on image classification.

**Adaptive labeled data acquisition**. We perform an ablation study by removing adaptive labeled data acquisition from MetaST (denoted as "MetaST w/o Labeled Data Acq."). Removing this component leads to around 2% performance drop on an average demonstrating the impact of labeled data acquisition. Moreover, the performance drop on SNIPS (39 slots) is larger than that on CoNLL03 (4 slots). This demonstrates that adaptive acquisition is more helpful for tasks with more slot types – where diversity and data distribution necessitate a better exploration strategy in contrast to random sampling employed in prior meta-learning works.

**Re-weighting strategies**. To explore the role of token-level re-weighting for pseudo-labeled sequences (discussed in Section 3.2), we replace our meta-learning component with different sample selection strategies based on the model confidence for different tokens. One sampling strategy chooses samples uniformly without any re-weighting (referred to as "MetaST w/o Re-weighting"). The sampling strategy with weights proportional to the model confidence favors easy samples (referred to as "MetaST-Easy"), whereas the converse favors difficult ones (referred to as "MetaST-Difficult").We observe the meta-learning based re-weighting strategy to perform the best. Interestingly, MetaST-Easy outperforms MetaST-Difficult significantly on CoNLL03 (EN) but achieves slightly lower performance on SNIPS. This demonstrates that difficult samples are more helpful when the quality of pseudo-labeled data is relatively high. On the converse, the sample selection strategy focusing on difficult samples introduces noisy examples with lower pseudo-label quality. Therefore, sampling strategies may need to vary for different datasets, thereby, demonstrating the necessity of adaptive data re-weighting as in our framework MetaST. Moreover, MetaST significantly outperforms classic self-training strategies with hard and soft pseudo-labels demonstrating the effectiveness of our design.

**Analysis of pseudo-labeled data re-weighting.** To visually explore the adaptive re-weighting mechanism, we illustrate token-level re-weighting of MetaST on CoNLL03 (EN) dataset with K=10 shot at step 100 in Fig. 2. We include the re-weighting visualisation on SNIPS in Appendix A.1. We observe that the selection mechanism filters out most of the noisy pseudo-labels (colored in blue) even those with high teacher confidence as shown in Fig. 2.

## 5 RELATED WORK

**Semi-supervised learning** has been widely used for consistency training (Bachman et al., 2014; Rasmus et al., 2015; Laine & Aila, 2017; Tarvainen & Valpola, 2017; Miyato et al., 2018), latent variable models (Kingma et al., 2014) for sentence compression (Miao & Blunsom, 2016) and code generation (Yin et al., 2018). More recently, methods like UDA (Xie et al., 2019) leverage consistency training for few-shot learning of instance-classification tasks leveraging auxiliary resources like paraphrasing and back-translation (BT) (Sennrich et al., 2016).

**Sample selection.** Curriculum learning (Bengio et al., 2009) techniques are based on the idea of learning easier aspects of the task *first* followed by the more complex ones. Prior work leveraging self-paced learning (Kumar et al., 2010) and more recently self-paced co-training (Ma et al., 2017) leverage teacher confidence to select easy samples during training. Sample selection for image classification tasks have been explored in recent works with meta-learning (Ren et al., 2018; Li et al., 2019) and active learning (Panagiota Mastoropoulou, 2019; Chang et al., 2017b). However, all of these techniques rely on only the model outputs applied to instance-level classification tasks.

**Semi-supervised sequence labeling.** Miller et al. (2004); Peters et al. (2017) leverage large amounts of unlabeled data to improve token representation for sequence labeling tasks. Another line of research introduces latent variable modeling (Chen et al., 2019; Zhou & Neubig, 2017) to learn interpretable and structured latent representations. Recently, adversarial training based model SeqVAT (Chen et al., 2020) and cross-view training method CVT (Clark et al., 2018) have shown promising results for sequence labeling tasks.

## 6 CONCLUSIONS

In this work, we develop an adaptive self-training framework MetaST that leverages self-training and meta-learning for few-shot training of neural sequence taggers. We address the issue of error propagation from noisy pseudo-labels from the teacher in the self-training framework by adaptive sample selection and re-weighting with meta-learning. Extensive experiments on six benchmark datasets and different tasks including multilingual NER and slot tagging for task-oriented dialog systems demonstrate the effectiveness of the proposed method particularly for low-resource settings.

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

# A APPENDIX

## A.1 EXPLORATIONS ON UNLABELED DATA AND MINI-BATCH S

**Variation in model performance with unlabeled data.** Table 12 shows the improvement in model performance as we inject more unlabeled data with diminishing returns after a certain point.

**Variation in model performance with mini-batch S.** We set the value of $S$ in Eq. 8 to $\{1, 3, 5\}$ respectively to explore its impact on the re-weighting mechanism. From Figure 3 we observe that the model is not super sensitive to hyper-parameter $S$ but can achieve a better estimate of the weights of the pseudo-labeled data with increasing mini-batch values.

| Ratio of Unlabeled Data | Datasets | |
|---|---|---|
| | SNIPS | CoNLL03 |
| 5% | 84.47 | 72.92 |
| 25% | 87.10 | 76.46 |
| 75% | 87.50 | 76.56 |

Table 7: Varying proportion of unlabeled data for MetaST with 10 labels per slot.

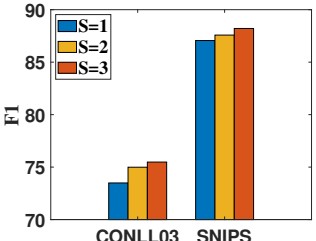

Figure 3: Varying $S$ mini-batch labeled data for re-weighting.

## A.2 ANALYSIS OF RE-WEIGHTING ON SNIPS AND CoNLL03

**Analysis of pseudo-labeled data re-weighting.** To visually explore the adaptive re-weighting mechanism, we illustrate token re-weighting of MetaST on CoNLL03 and SNIPS datasets with K=10 shot at step 100 in Fig. 4. Besides the observation in the experimental section, we observe that many difficult and correct pseudo-labeled samples (low teacher confidence) are selected according to Fig. 4a.

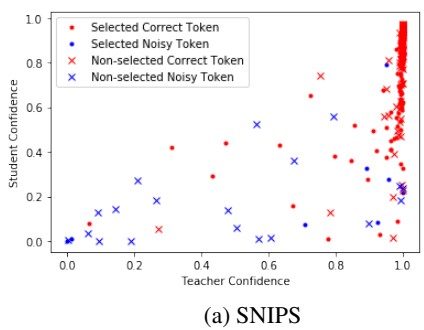 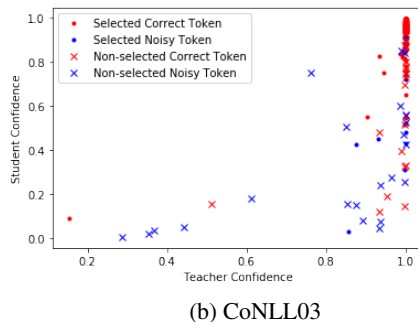

(a) SNIPS                    (b) CoNLL03

Figure 4: Visualization of MetaST re-weighting examples on SNIPS and CoNLL03 (EN).

## A.3 K-SHOTS

**Effect of varying the number of few-shots K.** We show the performance changes with respect to varying number of few-shots K $\{5, 10, 20, 100\}$ on Wikiann (en), MIT movie, MIT Restaurant, CoNLL2003 (En), Multilingual CoNLL and Multilingual Wikiann in Table 9-13. Since the number of labeled examples for some slots in Email dataset is around 10, we only show 5 and 10 shots for Email dataset in Table 8.

Table 8: Email Dataset.

| Method | Shots | |
|--------|-------|--------|
| | 5 | 10 |
| **Full-supervision** | | |
| BERT | | 0.9444 |
| **Few-shot Supervision** | | |
| BERT | 0.8211 | 0.8785 |
| **Few-shot Supervision + unlabeled data** | | |
| CVT | 67.44 | 78.24 |
| SeqVAT | 64.67 | 72.65 |
| Mean Teacher | 84.10 | 89.53 |
| VAT | 83.24 | 89.71 |
| Classic ST | 86.88 | 90.70 |
| BOND | 84.92 | 89.75 |
| MetaST | 89.21 | 92.18 |

| Method | Shots (3 Slot Types) | | | |
|--------|------|------|------|------|
| | 5 | 10 | 20 | 100 |
| **Full-supervision** | | | | |
| BERT | | 84.04 | | |
| **Few-shot Supervision** | | | | |
| BERT | 37.01 | 45.61 | 54.53 | 67.87 |
| **Few-shot Supervision + unlabeled data** | | | | |
| CVT | 16.05 | 27.89 | 46.42 | 66.36 |
| SeqVAT | 21.11 | 35.16 | 42.26 | 62.37 |
| Mean Teacher | 30.92 | 41.43 | 50.61 | 67.16 |
| VAT | 24.72 | 38.81 | 50.15 | 66.31 |
| Classic ST | 32.72 | 46.15 | 54.41 | 68.64 |
| BOND | 34.22 | 48.73 | 52.45 | 68.89 |
| MetaST | 55.04 | 56.61 | 60.38 | 73.20 |

Table 9: Wikiann (En) Dataset.

| Method | Shots (12 Slot Types) | | | |
|--------|------|------|------|------|
| | 5 | 10 | 20 | 100 |
| **Full-supervision** | | | | |
| BERT | | 87.87 | | |
| **Few-shot Supervision** | | | | |
| BERT | 62.80 | 69.50 | 75.81 | 82.49 |
| **Few-shot Supervision + unlabeled data** | | | | |
| CVT | 57.48 | 62.73 | 70.20 | 81.82 |
| SeqVAT | 60.94 | 67.10 | 74.15 | 82.73 |
| Mean Teacher | 58.92 | 67.62 | 75.24 | 82.20 |
| VAT | 60.75 | 70.17 | 75.41 | 82.39 |
| Classic ST | 63.39 | 71.88 | 76.58 | 83.06 |
| BOND | 62.50 | 70.91 | 75.52 | 82.65 |
| MetaST | 72.57 | 77.67 | 80.33 | 84.35 |

Figure 5: MIT Movie Dataset.

| Method | Shots (8 Slot Types) | | | |
|--------|------|------|------|------|
| | 5 | 10 | 20 | 100 |
| **Full-supervision** | | | | |
| BERT | | 78.95 | | |
| **Few-shot Supervision** | | | | |
| BERT | 41.39 | 54.06 | 60.12 | 72.24 |
| **Few-shot Supervision + unlabeled data** | | | | |
| CVT | 33.74 | 42.57 | 51.33 | 70.84 |
| SeqVAT | 41.94 | 51.55 | 56.15 | 71.39 |
| Mean Teacher | 40.37 | 51.75 | 57.34 | 72.40 |
| VAT | 41.29 | 53.34 | 59.68 | 72.65 |
| Classic ST | 44.35 | 56.80 | 60.28 | 73.13 |
| BOND | 43.01 | 55.78 | 59.96 | 73.60 |
| MetaST | 53.02 | 63.83 | 67.86 | 75.25 |

Table 10: MIT Restaurant Dataset.

| Method | Shots (4 Slot Types) | | | |
|--------|------|------|------|------|
| | 5 | 10 | 20 | 100 |
| **Full-supervision** | | | | |
| BERT | | 92.40 | | |
| **Few-shot Supervision** | | | | |
| BERT | 63.87 | 71.15 | 73.57 | 84.36 |
| **Few-shot Supervision + unlabeled data** | | | | |
| CVT | 51.15 | 54.31 | 66.11 | 81.99 |
| SeqVAT | 58.02 | 67.21 | 74.15 | 82.20 |
| Mean Teacher | 59.04 | 68.67 | 72.62 | 84.17 |
| VAT | 57.03 | 65.03 | 72.69 | 84.43 |
| Classic ST | 64.04 | 70.99 | 74.65 | 84.93 |
| BOND | 62.52 | 69.56 | 74.19 | 83.87 |
| MetaST | 71.49 | 76.65 | 78.54 | 85.77 |

Table 11: CoNLL2003 (EN)

| Method | Shots (4 Slot Types) | | | |
|--------|------|------|------|------|
| | 5 | 10 | 20 | 100 |
| **Full-supervision** | | | | |
| BERT | | 87.67 | | |
| **Few-shot Supervision** | | | | |
| BERT | 64.80 | 70.77 | 73.89 | 80.61 |
| **Few-shot Supervision + unlabeled data** | | | | |
| Mean Teacher | 64.55 | 68.34 | 73.87 | 79.21 |
| VAT | 64.97 | 67.63 | 74.26 | 80.70 |
| Classic ST | 67.95 | 72.69 | 73.79 | 81.82 |
| BOND | 69.42 | 72.79 | 76.02 | 80.62 |
| MetaST | 73.34 | 76.65 | 77.01 | 82.11 |

Table 12: Multilingual CoNLL03.

| Method | Shots (3 Slot Types × 41 languages) | | | |
|--------|------|------|------|------|
| | 5 | 10 | 20 | 100 |
| **Full-supervision** | | | | |
| BERT | | 87.17 | | |
| **Few-shot Supervision** | | | | |
| BERT | 77.68 | 79.67 | 82.33 | 85.70 |
| **Few-shot Supervision + unlabeled data** | | | | |
| Mean Teacher | 77.09 | 80.23 | 82.19 | 85.34 |
| VAT | 74.71 | 78.82 | 82.60 | 85.82 |
| Classic ST | 76.73 | 80.24 | 82.39 | 86.08 |
| BOND | 78.81 | 79.57 | 82.19 | 86.14 |
| MetaST | 79.10 | 81.61 | 83.14 | 85.57 |

Table 13: Multilingual Wikiann

## A.4 IMPLEMENTATIONS AND HYPER-PARAMETER

We do not perform any hyper-parameter tuning for different datasets. The batch size and maximum sequence length varies due to data characteristics and are as shown in Tbale 14. The hyper-parameters are as shown in Table 14.

Also, we retain parameters from original BERT implementation from `https://github.com/huggingface/transformers`.

We implement SeqVAT based on `https://github.com/jiesutd/NCRFpp` and implement CVT following `https://github.com/tensorflow/models/tree/master/research/cvt_text`.

| Dataset | Sequence Length | Batch Size | Labeled data sample size $|\mathcal{B}|$ | Unlabeled Batch Size | BERT Encoder |
|---|---|---|---|---|---|
| SNIPS | 64 | 16 | 32 | 32 | BERT-base-uncased |
| Email | 64 | 16 | 32 | 32 | BERT-base-cased |
| Movie | 64 | 16 | 32 | 32 | BERT-base-uncased |
| Restaurant | 64 | 16 | 16 | 32 | BERT-base-uncased |
| CoNLL03 (EN) | 128 | 16 | 8 | 32 | BERT-base-cased |
| Wikiann (EN) | 128 | 16 | 8 | 32 | BERT-base-cased |
| CoNLL03 (multilingual) | 128 | 16 | 32 | 32 | BERT-multilingual-base-cased |
| Wikiann (multilingaul) | 128 | 16 | 32 | 32 | BERT-multilingual-base-cased |

Table 14: Batch size, sequence length and BERT encoder choices across datasets

| | |
|---|---|
| BERT attention dropout | 0.3 |
| BERT hidden dropout | 0.3 |
| Latest Iteration R in labeled data acquisition | 5 |
| BERT output hidden size $h$ | 768 |
| Steps for fine-tuning teacher model on labeled data | 2000 |
| Steps T for self-training model on unlabeled data | 3000 |
| Mini-batch S | 5 |
| Re-initialize Student | Y |
| Pseudo-label Type | Hard |
| Warmup steps | 20 |
| learning rate $\alpha$ | $5e^{-5}$ |
| Weight_decay | $5e^{-6}$ |

Table 15: Hyper-parameters.

