# OpenReview forum: "Adaptive Self-training for Neural Sequence Labeling with Few Labels"
_ICLR.cc/2021/Conference — Reject_

### Official Review · AnonReviewer1 · 2020-10-29
**A self-learning framework for sequence labeling with solid experimental results**

**Rating:** 7
**Confidence:** 3

**Review:**

Summary

This paper proposes an adaptive self-training framework, called MetaST, for tackling few-shot sequence labeling tasks. The framework consists of several components: a teacher model that finetunes with the few-shot training data and generates noisy labels for the unlabeled examples; a student model that learns from re-weighted noisy labels (at the token level), and an iterative process to update the teacher with the trained student. It also uses a meta-learning mechanism to adjust the token-level weights based on a subsampled set of clean data. This subset is sampled based on the student model’s uncertainty to improve learning efficiency.

The proposed system is evaluated on a few sequence tagging tasks for slot filling or named entity recognition. It outperforms previous semi-supervised learning systems across all the evaluated tasks.

Strengths

- Very solid experimental results on both English and multilingual datasets. The comparison against previous systems (including ones using BERT, similar to the proposed model) seems quite thorough.
- Ablation studies that showcase the effectiveness of each model component.

Weaknesses

- The framework is quite complex with many subcomponents: uncertainty-based data acquisition, meta-learning based token-level re-weighting, iterative updates, etc. It is nice that the paper contains a fairly complete ablation study to analyze the effectiveness of each of these components, though.

Other questions/comments:

- In Algorithm 1, how do you check convergence? How many steps does that usually take?
- In the last paragraph of section 4, “Analysis of pseudo-labeled data re-weighting”, does “step 100” refer to the step of the student model or the outer loop (teacher re-initialization)?

---

> ### Author Response · Authors · 2020-11-14
> **Author Response to Reviewer 1**
>
> Thanks a lot for your constructive feedback and acknowledgement of our extensive experiments and ablation studies. To the best of our knowledge, this is the first work to extensively study few-shot neural sequence labeling with significant improvements over state-of-the-art baselines on several benchmark datasets including multiple shots, slots, domains, and languages. We summarize some additional findings in our experiments complementing those outlined in previous responses.
>
> * __The necessity of adaptive data re-weighting__. In Table 6, MetaST-Easy sampling strategy outperforms MetaST-Difficult one significantly on CoNLL03(EN) but achieves slightly lower performance on SNIPS.  This depicts that sampling strategies may need to vary for different datasets, thereby demonstrating the necessity of adaptive data re-weighting as in our framework MetaST. Similar observation has been noted in prior work [1] in the context of self-training under domain shift.
> * __Soft labels v.s. Hard labels.__ In Table 6, while replacing hard pseudo-labels with soft ones (denoted as “MetaST w/ Soft Pseudo-Labels”), we observe performance drop on SNIPS and CoNLL(EN). This suggests hard or sharpened labels are more informative than soft ones especially with an effective data selection strategy. Similar observation has been noted in prior work [2] in the context of self-training for image classification.
>
> > _1. “In Algorithm 1, how do you check convergence? How many steps does that usually take?”_
>
> In this work, we train for 3000 steps per self-training iteration with 20000 as max-steps overall. Based on our observations, the models converge to stable performance within the max-steps.
>
> > _2. ”In the last paragraph of section 4, “Analysis of pseudo-labeled data re-weighting”, does “step 100” refer to the step of the student model or the outer loop (teacher re-initialization)?”_
>
> Sorry for the confusion. “Step 100” here refers to the step of the student model. We will make it clear in the revised version.
>
> [1] Sebastian Ruder, Barbara Plank. Strong Baselines for Neural Semi-supervised Learning under Domain Shift. ACL 2018
>
> [2] Ananya Kumar, Tengyu Ma, Percy Liang. Understanding Self-Training for Gradual Domain Adaptation ICML 2020

---

### Official Review · AnonReviewer3 · 2020-10-30
**ablation study missing**

**Rating:** 7
**Confidence:** 2

**Review:**

SUMMARY
The paper presents a series of strategies for self-supervised learning for sequence labeling tasks.
The proposed model is a teacher-student network.
A teacher model is trained on a small set of labeled data, and then is used to pseudo-annotate a lot of unlabeled data. The student model is trained on the pseudo-labeled data.
This paper introduces a strategy to select informative labeled examples to use as dev set for the student model, and adapts an existing re-weighting mechanism for pseudo-labeled examples to the sequence labeling setting.

The overall approach is tested on different sequence labeling datasets with different characteristics.
In low resources scenarios, the proposed model significantly outperforms previous models.

------------------------
REVIEW

Please, make clear that the model is thought for a low resource setting.
It would be interesting to see an experiment where the entire dataset is used for training the teacher model and an external unlabeled data is pseudo labeled.
Following the experiments in table 5, this should not work as good as in a low resource.
The authors should give a intuition, or an answer on why this is the case.

There is an important experiment missing.
The impact of adaptive label data acquisition is not tested.
What happens if you keep all the examples?

The argument in the introduction about self learning not suitable for sequence learning model is a bit weak. Self learning has been extensively studied and successfully applied in sequence labeling tasks (for example, https://arxiv.org/pdf/1804.09530.pdf).

Why do you need to sum in equation 8? do you have same instances and same tokens in different batches? why do you need that?
In general section 3.2 could be a bit  more clear.

- Presentation

Some sentences are very long and hard to read, i.e., "To address such issues stemming from noisy labels and training set biases, learning to re-weight noisy examples (Ren et al., 2018) lever- ages a meta objective with the basic assumption that the best weighting strategy should minimize the loss on a held-out clean labeled validation set."

Ren et al. 2018 has been published at ICML 2018, please update the references.


-- UPDATE
Thanks for the clear and exhaustive response.
Minor, regarding the ablation study in A.1, with S=3 you get the best results, why not try with more?

---

> ### Author Response · Authors · 2020-11-14
> **Author Response to Reviewer 3 (Part 1)**
>
> Thanks for your constructive review and we are glad to incorporate your suggestions to improve our paper. Please refer to the general response for the task setup and the key objective.
>
> In this paper, we study the problem of sequence labeling with few annotated/labeled examples for each slot. More specifically, we leverage a small set of labeled data and a large amount of in-domain unlabeled data to improve training efficiency for a given task. While prior work in transfer learning and domain shift / adaptation (e.g. adapting tasks from a high resource source domain to a low resource target domain) are related to our problem, this is not our focus. Our setting is useful for scenarios where it is difficult to obtain large-scale manual annotations for a task either due to cost or compliance concerns (e.g., task-oriented dialog systems with sensitive user information that we cannot manually access and annotate). To the best of our knowledge, this is the first work to develop a few-shot neural sequence labeling model to obtain significant performance improvements over several state-of-the-art baselines over a wide range of tasks with multiple shots, slots, domains, and languages. We provide detailed responses to other concerns as follows.
>
> > _1. “Please, make clear that the model is thought for a low resource setting. It would be interesting to see an experiment where the entire dataset is used for training the teacher model and an external unlabeled data is pseudo labeled. Following the experiments in table 5, this should not work as good as in a low resource. The authors should give a intuition, or an answer on why this is the case.”_
>
> Thanks for your constructive suggestions. Methods like self-training and meta-learning have been shown to be more useful for tasks with limited labeled data, and the impact/improvement decreases with an increasing amount of labeled examples [2]. Particularly, for self-training, given the ceiling performance for every task and the improved performance of the teacher with more training labels, there is less room for (relative) improvement for the student over the teacher model. Consider the SNIPS dataset for example. Our model obtains 12% and 2% improvement over the few-shot BERT model for the 10-shot and 100-shot setting with F1-scores as 88.22% and 95.39%, respectively. The ceiling performance for this task is 95.8% on training BERT on the entire dataset with 13K labeled examples. Please note that there is no additional unlabeled data over that reported in Table 1. For each task/dataset, we randomly sampled K labeled examples from the corresponding Train set in Table 1, while treating the remaining as in-domain unlabeled data by disregarding their labels -- following standard setups for semi-supervised learning. We repeatedly sample K labeled instances three times for multiple runs and report average performance with standard deviation across the runs.
>
> > 2. _“There is an important experiment missing. The impact of adaptive label data acquisition is not tested. What happens if you keep all the examples?”_
>
> **We have reported this important ablation study result in Table 6**. We remove the adaptive labeled data acquisition from MetaST (denoted as “MetaST w/o Labeled Data Acq.”) that leads to around 2% performance drop on average. To emphasize the importance of these results, we will add one corresponding analysis in the experimental section for better illustration. For additional discussions regarding adaptive data acquisition, please refer to our corresponding response to Reviewer 2.

---

> > ### Author Response · Authors · 2020-11-14
> > **Author Response to Reviewer 3 (Part 2)**
> >
> > > 3. _”The argument in the introduction about self learning not suitable for sequence learning model is a bit weak. Self learning has been extensively studied and successfully applied in sequence labeling tasks (for example, https://arxiv.org/pdf/1804.09530.pdf).”_
> >
> > Thanks a lot for pointing out this important reference paper on “Strong Baselines for Neural Semi-supervised Learning under Domain Shift” (https://arxiv.org/pdf/1804.09530.pdf) that supports many of our claims. The paper investigated bootstrapping approaches including self-training in the context of neural networks under domain shifts. Although our few-shot neural sequence labeling setting is different from theirs, we do arrive at similar conclusions:
> > * According to their experimental results on sequence labeling for POS tagging (Page 7), they conclude that “**self-training does not work for the sequence prediction task**”. Such an observation supports our argument in the introduction that self-training may not be directly applicable or may not yield the best results for sequence labeling tasks.
> > * The authors further note: “Its (self-training) main downside is that the model is not able to correct its own mistakes and errors are amplified, an effect that is increased under domain shift.” This is one of our main motivations for pseudo-labeled sample selection and re-weighting mechanism to mitigate error propagation in iterative self-training.
> > * They also mention “some success achieved with careful task-specific data selection (Petrov and McDonald, 2012), while others report limited success on a variety of NLP tasks.” In this work, we develop a method that works out-of-the-box without any manually designed task-specific strategies/assumptions. We demonstrate this via extensive experiments on a wide benchmark of tasks with multiple shots, slots, domains, and languages.
> >
> > > 4. _”Why do you need to sum in equation 8? do you have same instances and same tokens in different batches? why do you need that? In general section 3.2 could be a bit more clear.”_
> >
> > We sample S different batches of labeled data leveraging the adaptive data acquisition strategy and calculate the mean of the gradients from Equation 7. Note that S is a constant number that is the same for each token and the proportional sign in Equation 8: $w_{m, n} \propto  \max (\sum_{s=1}^S u_{m, n, s}, 0)$. Sequence labeling tasks have inter-dependencies in the token- and corresponding label-space. Therefore, increasing the number of batches help in better weight estimation. We conduct an ablation study of S and show its results in the Appendix A.1.
> >
> >
> > > 5. _“Some sentences are very long and hard to read, i.e., "To address such issues stemming from noisy labels and training set biases, learning to re-weight noisy examples (Ren et al., 2018) lever- ages a meta objective with the basic assumption that the best weighting strategy should minimize the loss on a held-out clean labeled validation set." Ren et al. 2018 has been published at ICML 2018, please update the references.”_
> >
> > Thanks for your helpful suggestion. We will paraphrase the long sentences and update the corresponding reference as suggested.
> >
> > [1] Sebastian Ruder, Barbara Plank. Strong Baselines for Neural Semi-supervised Learning under Domain Shift. ACL 2018
> >
> > [2] Barret Zoph, Golnaz Ghiasi, Tsung-Yi Lin, Yin Cui, Hanxiao Liu, Ekin Cubuk, Quoc V. Le. Rethinking Pre-training and Self-training. NeurIPS 2020

---

### Official Review · AnonReviewer2 · 2020-11-02
**The limited novelty, lack of sufficient rigor around proposals make it less publish-worthy.**

**Rating:** 4
**Confidence:** 4

**Review:**

The authors propose adaptive self-training that uses self-training + meta-learning for few-shot training of neural sequence taggers. Specifically the authors focus on reducing noisy training data for student models and reweighting them.


ADAPTIVE LABELED DATA ACQUISITION:
* The motivation was not clear to me.
* The authors proposed a way to select example based on teacher loss, but it's unclear to me why this was done in this way. I did not find sufficient discussion around this. What happens if we don't do this?
* Overall, It would've been nice to see a deeper discussion around whether this is needed in the first place, what benefits does it give, and if so what are all the ways of solving this problem and why the particular approach is the right one.

RE-WEIGHTING PSEUDO-LABELED DATA
* I could not find sufficient novelty about the "token" aspect. What's special about perturbing weight for each token vs instance.
* the entire section was a bit unconvincing, I could not find the motivation for diversity, nor sufficient rigor for the choices made.


Experimental Results
* It's nice to see several experiments.
* Have the authors considered introducing a pretraining objective on the unlabeled data? I wonder how much of the "gap" from pretrain+finetune would go away if the teacher models were pretrained on the in-domain unlabeled data.
* Have the authors compared with an explicit distillation step of a teacher (E.g. BERT+finetune + distill)? I am asking because this might really show whether we need 'reweighting', noise reduction etc.

---

> ### Author Response · Authors · 2020-11-14
> **Author Response to Reviewer 2 (Part 1)**
>
> We want to thank the reviewer for thoughtful comments and we would be glad to incorporate your suggestions to improve our paper. Please refer to the general response for the task setup and the key objective.
>
> In this paper, we focus on the problem of sequence labeling with few annotated examples (e.g., K=10) for each slot for each task/dataset. This few shot setting is particularly challenging for structured prediction tasks, where self-training has not been effective in the past [1]. The main challenge is error propagation due to noisy pseudo-labels over iterative training. We address this error propagation/amplification by leveraging meta-learning with adaptive labeled data acquisition and re-weighting pseudo-labeled data as follows.
>
> > ADAPTIVE LABELED DATA ACQUISITION :
> > 1. _“Overall, It would've been nice to see a deeper discussion around whether this is needed in the first place, what benefits does it give, and if so what are all the ways of solving this problem and why the particular approach is the right one.”_
>
> **Why do we need labeled data acquisition?** Given N slots for each task and K labels for each slot, the model has access to K\*N labeled examples. The role of labeled data acquisition is to construct an informative held-out validation set from K\*N labeled examples for the pseudo-labeled data re-weighting mechanism. This held-out data is necessary for the meta-learning setup. Previous work on using meta-learning for classification tasks [2, 3] constructs this validation set by random sampling. In contrast to instance-level classification tasks, sequence labeling tasks require labels for each token in the sequence where the tokens and correspondingly its labels have inter-dependencies. Therefore this validation set generation can benefit from a better exploration strategy instead of random sampling.
>
> **Our labeled data acquisition strategy:** We leverage uncertainty sampling to select samples that the model is confused about and correspondingly can benefit from knowing their labels (similar strategies have been used in active learning settings). We use stochastic loss decay from the model as a proxy for the model uncertainty to generate validation set on the fly --- that is used for weight-estimation and re-weighting pseudo labeled data in the next step. We empirically demonstrate its impact via ablation study, where MetaST w/o Labeled Data Acq” in Table 6 demonstrates 2% average degradation on replacing this component by random sampling in MetaST.
>
> **Other acquisition strategies:** Data acquisition strategies can be divided into random [2,3], easy [4] and hard example mining [5] or uncertainty-based methods [6].  Easy and hard example mining methods work well in different scenarios, where prior works [6, 12] show uncertainty-based methods to have better generalizability across diverse settings. There are several approaches to uncertainty estimation including error decay [13, 14], Monte Carlo dropouts [12] and predictive variance [6].
>
> > RE-WEIGHTING PSEUDO-LABELED DATA:
>
> > 2. _“I could not find sufficient novelty about the "token" aspect. What's special about perturbing weight for each token vs instance. The entire section was a bit unconvincing, I could not find the motivation for diversity, nor sufficient rigor for the choices made.”_
>
> **Token-level vs. instance-level re-weighting:** The teacher model assigns pseudo-labels to each token in a sequence. Since token-level pseudo-labels are noisy, we want to re-weight them. Instance-level re-weighting assumes the quality of all token-level pseudo-labels in the sequence to be similar which is not valid given the variable performance of the model for different slot types.
>
> **Token-level weight perturbation** follows a similar principle to that of an instance. However, the key challenge is to construct an informative held-out validation set for the re-weighting procedure. This is where the adaptive labeled data acquisition strategy from the previous step comes into play. This selects the most informative (given by model uncertainty) labeled data samples to construct a small held-out validation set. The token weights are estimated based on the loss changes on this validation set.

---

> > ### Author Response · Authors · 2020-11-14
> > **Author Response to Reviewer 2 (Part 2) Cont'd**
> >
> > **Diversity of slot types:** Sequence labeling tasks involve many slots (e.g. WikiAnn has 123 slots) with variable difficulty and distribution in the data. For example, CoNLL03 (EN) has four slot types, where MISC is used to denote all other entity types besides PER, LOC and ORG. In case of naive sampling, the model would oversample from the most populous category. This is even more critical in the multilingual setting with a variable amount of unlabeled data per language, where the low-resource languages and corresponding slot types may suffer if we do not emphasize on sampling diverse representatives across different slot types and languages. This is yet another motivation for judiciously constructing the held-out validation set by the adaptive acquisition strategy instead of random selection.
> >
> > **Motivation behind weight perturbation:** Weight perturbation is used to discover data points that are most important to improve the model performance on a held-out validation set [7]. The higher the negative gradients, the more important are the data samples [7]. We set the weights to be proportional to the negative gradients (Eq. 7 misses a negative sign, we will fix this typo in the revision) to reflect the importance of pseudo-labeled samples. Such technique has been shown to be useful for instance-level classification tasks in [2]. Considering the challenge of sequence labeling tasks, we sample S batches of labeled data from the adaptive acquisition strategy and calculate the mean of the gradients. Note that S is a constant number that is the same for each token and the proportional sign in Equation 8. Since a negative weight indicates a pseudo-label of poor quality that would potentially degrade the model performance, we set such weights to 0 to filter them out.
> >
> > > Experimental Results
> >
> > > 3.  _”Have the authors considered introducing a pretraining objective on the unlabeled data? I wonder how much of the "gap" from pretrain+finetune would go away if the teacher models were pretrained on the in-domain unlabeled data.”_
> >
> > We performed the following experiments as suggested. The BERT model is further pre-trained on in-domain unlabeled data and then fine-tuned with the few labeled data denoted as “BERT (Continual Pre-training + Few-shot Supervision)”. The results are reported on SNIPS and CoNLL03(EN) with 10 shots. The pre-training step improves the BERT performance over the baseline on SNIPS but degrades the performance on CoNLL03. This indicates that continual pre-training can improve the performance of few-shot supervised BERT on specialized tasks (e.g., SNIPS) with different data distribution than the original pre-training data (e.g., Wikipedia), but may not help for general domain ones like CoNLL03 with overlapping data from Wikipedia. In paper [8], the authors conduct an extensive study on contrasting self-training and pre-training for image classification tasks and observe that self-training has more flexibility as well as effectiveness in both low-data and high-data regime. Interestingly, our continual pre-training results are quite similar to the classic self-training ones on these datasets (Classic ST in Table 2), thereby, demonstrating the need for better pseudo-labeled sample selection and re-weighting to mitigate the noise in self-training. In contrast to the above baselines, MetaST brings significant improvements on both datasets.
> >
> > __Results:__
> >
> > | Models           |       SNIPS |       CoNLL03 |
> > | :---  | :------:  | :------:  |
> > | BERT (Few-shot Supervision)      |    79.01         |      71.15|
> > | BERT ( Continual Pre-training + Few-shot Supervision)   |  83.96        |  69.84|
> > | Classic ST   |  83.26        |  70.99 |
> > | MetaST(ours)    |  88.22        |          76.65|
> >
> > > Experimental Results
> >
> > > 4. _“Have the authors compared with an explicit distillation step of a teacher (E.g. BERT+finetune + distill)? I am asking because this might really show whether we need 'reweighting', noise reduction etc.”_
> >
> > Distillation is related to self-training with some key differences. During distillation [9, 10, 11], the student is trained to match the performance of the teacher, where the teacher model is typically frozen with fixed parameters. But in self-training, the teacher and student model share parameters and both of them are iteratively updated to improve each other and the pseudo-labeled data quality. In view of these standard setups, can you please elaborate on an “explicit distillation step of a teacher”?

---

> > > ### Author Response · Authors · 2020-11-14
> > > **Author Response to Reviewer 2 (Part 3) Cont'd**
> > >
> > > Based on our understanding of explicit distillation, we trained a student model with two losses including fine-tuning loss with few-shot supervision and distillation loss with soft labels from the teacher.  We report experimental results with the following settings: (1) distillation with fixed teacher model denoted as BERT (Few-shot Supervision + Distill),  and (2) self-training by replacing the teacher with a well-trained student denoted as “Self-training with soft labels”  to answer the above question and analyze the necessity of noise reduction.
> > >
> > > * We conduct experiments on SNIPS and CoNLL03 with the 10-shot setting. Experimental results show that BERT with few-shot supervision and distillation loss slightly outperform BERT with few-shot supervision. However, the performance is lower than that of the self-training setting demonstrating the effectiveness of the self-training mechanism. However, our proposed framework significantly outperforms these two baselines as well as “self-training with soft labels” on both datasets.
> > >
> > > * To explicitly analyze the impact of re-weighting, we had conducted an ablation study, i.e. “MetaST w/o Re-weighting” with results on SNIPS and CoNLL03 in Table 6. Removing the re-weighting mechanism leads to an average of 3% performance drop on both datasets.
> > >
> > > * As discussed in [1], the main drawback of self-training is that the model cannot correct the noisy predictions and further amplifies the errors. While some success has been achieved with careful task-specific data selection [2], it is difficult to generalize for arbitrary tasks. Our proposed adaptive self-training method works out-of-the-box without task-specific assumptions and tested on 6 benchmark datasets with multiple shots, slot types, domains and languages -- where it has consistently outperformed several state-of-the-art baselines.
> > >
> > > **Results:**
> > >
> > > | Models           |       SNIPS |       CoNLL03 |
> > > | :---  | :------:  | :------:  |
> > > | BERT (Few-shot Supervision)            |    79.01         |      71.15|
> > > | BERT (Few-shot Supervision + Distill)     | 79.97         |  71.65|
> > > | Self-training with soft labels       | 81.17          |   71.87|
> > > | MetaST(ours)    |  88.22        |          76.65|
> > >
> > > [1] Sebastian Ruder, Barbara Plank. Strong Baselines for Neural Semi-supervised Learning under Domain Shift. ACL 2018
> > >
> > > [2] Mengye Ren, Wenyuan Zeng, Bin Yang, Raquel Urtasun. Learning to Reweight Examples for Robust Deep Learning. ICML 2018
> > >
> > > [3] Jun Shu, Qi Xie, Lixuan Yi, Qian Zhao, Sanping Zhou, Zongben Xu, Deyu Meng. Meta-Weight-Net: Learning an Explicit Mapping For Sample Weighting. NeurIPS 2019
> > >
> > > [4] M. P. Kumar, B. Packer, and D. Koller. Self-paced learning for latent variable models. In NIPS, 2010.
> > >
> > > [5] A. Shrivastava, A. Gupta, and R. Girshick. Training region-based object detectors with online hard example mining. In CVPR, 2016.
> > >
> > > [6] Haw-Shiuan Chang, Erik Learned-Miller, Andrew McCallum. Active Bias: Training More Accurate Neural Networks by Emphasizing High Variance Samples. NIPS 2017
> > >
> > > [7] Koh, Pang Wei and Liang, Percy. Understanding black-box predictions via influence functions.  ICML, 2017
> > >
> > > [8] Barret Zoph, Golnaz Ghiasi, Tsung-Yi Lin, Yin Cui, Hanxiao Liu, Ekin Cubuk, Quoc V. Le. Rethinking Pre-training and Self-training. NeurIPS 2020
> > >
> > > [9] Geoffrey Hinton, Oriol Vinyals, Jeff Dean. Distilling the Knowledge in a Neural Network. arXiv preprint arXiv:1503.02531
> > >
> > > [10] Raphael Tang, Yao Lu, Linqing Liu, Lili Mou, Olga Vechtomova, and Jimmy Lin.  Distilling task specific knowledge from BERT into simple neural networks. CoRR, abs/1903.12136.
> > >
> > > [11] Wenhui Wang, Furu Wei, Li Dong, Hangbo Bao, Nan Yang, Ming Zhou. MINILM: Deep Self-Attention Distillation for Task-Agnostic Compression of Pre-Trained Transformers. NeurIPS 2020
> > >
> > > [12] Yarin Gal, Riashat Islam, Zoubin Ghahramani. Deep Bayesian Active Learning with Image Data. ICML 2017
> > >
> > > [13] Konyushkova K, Sznitman R, Fua P Learning active learning from data. In: NIPS 2017
> > >
> > > [14] Haw-Shiuan Chang, Shankar Vembu, Sunil Mohan, Rheeya Uppaal & Andrew McCallum. Using Error Decay Prediction to Overcome Practical Issues of Deep Active Learning for Named Entity Recognition, Machine Learning, 109, 1749–1778, 2020

---

### Author Response · Authors · 2020-11-14
**Overall Summary and General Response for all Reviewers**

We thank all the reviewers for their constructive feedback. We are currently working on revising the paper to accommodate reviewer suggestions and will upload an updated version soon. In the meantime, we would like to clarify some confusions about our task setup / problem statement followed by more detailed responses.

**Task setup / problem statement:** In this work, we consider a few labeled/annotated examples (e.g., K=10) for each slot for each task (i.e. dataset in the context of our experiments) and large amounts of in-domain unlabeled data available for the given task. For example, for Named Entity Recognition (NER) task with 4 slots like PER, ORG, LOC and MISC -- there are 10\*4=40 labeled examples available for the corresponding dataset. We refer to this setup as the few-shot or K-shot setting. Additionally, we adopt the BIO tagging policy for every task, where each token is annotated with a span marker designating the beginning (B), intermediate (I) or outside (O) of the entity span. For example, the above NER task has 4\*2+1=9 classes (with ‘O’ shared between the slot types). Our objective is to assign each token in a sentence/sequence to one of those classes.

**Missing ablation experiment:** We do have this ablation experiment in the paper corresponding to “MetaST w/o Labeled Data Acq” in Table 6 that demonstrates 2% average degradation in removing this component from MetaST.

**Is this a solved problem?** Our paper has two main objectives: (i) Develop a model for few-shot sequence labeling following the above problem statement, and (ii) Leverage self-training for the same. Paper [1] (page 7) cited by Reviewer 3 postulates “Self-training does not work for the sequence prediction task in the low-data regime.… Its (self-training) main downside is that the model is not able to correct its own mistakes and errors are amplified”. This is one of our main motivations for pseudo-labeled sample selection and re-weighting mechanism to mitigate error propagation in iterative self-training. We show that self-training is useful in the context of few-shot sequence prediction and propose several enhancements with extensive ablation experiments (Table 6) to empirically support our claims and the impact of different components in our framework.

**Key result:** While there has been some success with careful task-specific data selection [2] and some NLP tasks [3, 4, 5], to the best of our knowledge, this is the first work on few-shot sequence labeling that shows large improvements over several benchmark datasets including user utterances from task-oriented dialog systems and massive multilingual NER (41 languages) with different slots, shots, domains and languages. Specifically, we show that for the 10-shot setting, our method MetaST works out-of-the-box without any task-specific assumptions obtaining 10% improvement over state-of-the-art systems.

We have provided more details about the motivations for adaptive labeled data acquisition and re-weighting pseudo-labeled data, and additional experiments to the individual reviewer responses. We will add these reviewer suggestions into the revised version and improve our paper presentation.

[1] Sebastian Ruder, Barbara Plank. Strong Baselines for Neural Semi-supervised Learning under Domain Shift. ACL 2018

[2] Slav Petrov and Ryan McDonald. Overview of the 2012 shared task on parsing the web. Notes of the First Workshop on Syntactic Analysis of NonCanonical Language (SANCL), 2012

[3] Barbara Plank. 2011. Domain adaptation for parsing. University Library Groningen.

[4] Vincent Van Asch and Walter Daelemans. Predicting the effectiveness of self-training: Application to sentiment classification. arXiv preprint arXiv:1601.03288

[5] Rob van der Goot, Barbara Plank, and Malvina Nissim.. To normalize, or not to normalize: The impact of normalization on part-of-speech tagging. In Proceedings of the 3rd Workshop on Noisy User generated Text. 2017

---

### Author Response · Authors · 2020-11-20
**Summary of changes**

We thank all the reviewers for their constructive feedback. We finished revising the paper and uploaded an updated version. The major changes are __colored in blue__ for reading convenience.  We believe that this version has better clarity and readability by incorporating the reviewers’ feedback. We summarize our changes below:
1. We added observations from paper [1] (as suggested by reviewer 3) as a motivation for our work to illustrate the challenges of applying self-training for sequence labeling tasks in a low-data regime in __Pages 1-2__.
2. We define the key challenges between instance-level and token-level re-weighting via meta-learning as well as corresponding motivations on __Page 2__.
3. We follow the suggestion of reviewer 2 to provide a deeper discussion about labeled data acquisition (motivation, existing techniques, alternatives, etc.) on __Page 4__.
4. We follow the suggestions of reviewer 2 and reviewer 3 to provide more motivation behind our choices in the re-weighting mechanism on __Page 5__.
5. Following the suggestion of reviewer 2, we provide an explanation about the decrease in relative improvement with more labeled data on __Page 8__.
6. Following the suggestion of reviewer 3, we added experiments about “continued pre-training” as well as “self-training with soft pseudo-labels” (as used in knowledge distillation with detailed explanations in author response) on __Page 8__ and __Table 6__.
7. While we had reported the ablation study for adaptive data acquisition in the submitted draft, we added additional analysis on __Page 9__ to address reviewer 3’s concerns.

Please let us know whether this revision addresses your concerns and if you have any other questions or comments. We are looking forward to a constructive discussion.

 [1] Sebastian Ruder, Barbara Plank. Strong Baselines for Neural Semi-supervised Learning under Domain Shift. ACL 2018

---

### Decision · Program_Chairs · 2021-01-07
**Final Decision**

**Decision:**

Reject

**Comment:**

This paper introduces a self-training strategy for semi-supervised learning for few shot sequence learning.   It builds on ideas from an existing work on robust deep learning that adaptively reweights examples for learning to reduce impact of noisy examples, here the noisy examples are introduced to the student network training by the teacher network.  Two main novel points, one is on  selectively constructing the validation set used for adaptive reweighting.  Another idea is to move from the sentence level reweighting to token level reweighting.   The paper shows strong results suggesting the proposed method can effectively learn under few-shot learning.
A primary concern from the reviewers is that the paper has limited novelty given that it primarily applies existing ideas to a slightly different problem.  Another concern is that the system consists of many components, each of the choices could have other viable options.  The ablation studies indicate these components are useful compared to when removed, but fail to explore possible alternative choices. One of the questions is whether token-level reweighting is necessary. It would have been nice to see an ablation study comparing against a baseline using sentence-level reweighting.